# Germanium Nanowires as Sensing Devices: Modelization of Electrical Properties

**DOI:** 10.3390/nano11020507

**Published:** 2021-02-17

**Authors:** Luca Seravalli, Claudio Ferrari, Matteo Bosi

**Affiliations:** IMEM-CNR (Institute of Materials for Electronics and Magnetism—Consiglio Nazionale delle Ricerche), Parco delle Scienze 37a, 43124 Parma, Italy; claudio.ferrari@imem.cnr.it (C.F.); matteo.bosi@imem.cnr.it (M.B.)

**Keywords:** germanium nanowires, nanosensors, sensing nanostructures, molecular functionalization, electrical properties simulation, modeling of carrier transport

## Abstract

In this paper, we model the electrical properties of germanium nanowires with a particular focus on physical mechanisms of electrical molecular sensing. We use the Tibercad software to solve the drift-diffusion equations in 3D and we validate the model against experimental data, considering a p-doped nanowire with surface traps. We simulate three different types of interactions: (1) Passivation of surface traps; (2) Additional surface charges; (3) Charge transfer from molecules to nanowires. By analyzing simulated I–V characteristics, we observe that: (i) the largest change in current occurs with negative charges on the surfaces; (ii) charge transfer provides relevant current changes only for very high values of additional doping; (iii) for certain values of additional n-doping ambipolar currents could be obtained. The results of these simulations highlight the complexity of the molecular sensing mechanism in nanowires, that depends not only on the NW parameters but also on the properties of the molecules. We expect that these findings will be valuable to extend the knowledge of molecular sensing by germanium nanowires, a fundamental step to develop novel sensors based on these nanostructures.

## 1. Introduction

Germanium nanowires (NWs) are one-dimensional semiconducting nanostructures that are currently being investigated for the development of innovative molecular sensors, among other nanodevices. Convincing and encouraging results have been already obtained by using these nanostructures to electrically detect various molecules [1,2,3,4].

Research efforts are currently focusing on understanding in depth the sensing mechanisms, in order to optimize the nanostructures for the development of high-performing electrical sensors. There is a consensus that surface traps in these nanostructures play a major role in determining the NW electrical properties [5,6,7]. For example, it was shown that molecular functionalization of NWs can not only passivate surface traps, but induce a change in the surface potential [7]. Moreover, different functionalization protocols have been put forward with different molecules that can induce different changes in surface potential state [8,9].

Hence, a fundamental question can be raised: what is the physical principle of operation for electrical sensing molecules with Ge NWs?

First of all, it has been now established that Ge NWs show a p-type behavior, because of an accumulation of holes at the surface owing to the effect of surface traps. [10] Therefore, three basic types of interaction between molecules and NWs can be expected to alter the NW electrical characteristics and should be considered:(1)Passivation of surface traps: for example, it has been shown that isoprene-treated NWs show a strong reduction of surface charges, [10] as thiol-based molecules allow for an improvement of electrical performances via passivation of surface states [11,12];(2)Effect of charges due to molecules adsorbed on surface: adsorbed water molecules on the surface of Si NWs lead to the formation of OH- ions, with a net negative charge on the surface [13];(3)Charge transfer from functionalized molecules to the NW, resulting in an effective doping of the surface, as obtained by a deposition of MoO_3_ thin film on Ge NWs [14]. Doping by surface charge transfer during growth is a known method for controlling the carrier density in NWs, [15] and the method of molecular layer doping has been proposed as a viable approach to control doping both in bulk Ge [16,17] and in Ge nanostructures [18,19]. N-doping of Si NWs due to attachment of terpyridine has been demonstrated almost 10 years ago [20].

It should be noticed that these processes may be acting at the same time, depending on the molecule involved in the sensing process: for example, it has been argued that in the case of water both charge transfer and field effect due to attached water dipoles could be responsible for a change of the resistance of Ge NWs [4].

From a theoretical perspective, up to now, models focused on the effect of surface traps’ presence and their passivation. On the other hand, less attention has been given to additional surface charges due to molecules, despite the fact that surface states are the fundamental element in determining the characteristic of sensing devices based on NWs [5,21]. Recently, it was demonstrated that, by electrostatic gating, it is even possible to change polarity of transport and obtain negative differential resistance via surface doping of NWs [6].

In this paper, we develop and study a theoretical model of the electrical properties of Ge NWs and we consider the effect of passivation, of additional surface charge of adsorbed molecules and of charge transfer from molecules to NWs, with particular attention to the modeling of the surface potential. For this, we solve the classical drift-diffusion differential equations, by relying on empirical parameters provided by experimental data and by previously published results. This approach allows for simple and fast calculations of NW electrical properties considering different molecules and different interactions.

The results of this modelization allow to understand in depth the effect of functionalization by molecules on the electrical properties of NW and, thus, the expected performance of this nanostructure as a sensor. For example, passivation is expected to increase the mobility, thanks to the reduction of surface states, while charge transfer could dictate the level of intrinsic doping needed for best operation.

We stress the fact that we aim to predict the maximum effect that can be expected upon a certain process: therefore, we consider the highest possible change in the involved parameters—for example, total passivation of all surface traps, n and p charge transfer up of 10^18^ cm^−3^, and surface charge accumulation up to a density of 10^12^ cm^−2^.

## 2. Model Details

For the modelization of the NW system and its electric properties we used TiberCAD 3.0 (Tiberlab SRL, Rome, Italy), a proven effective simulation software [22,23], that has been successfully used in the past to model properties of semiconductor low-dimensional nanostructures, such as quantum dots [22,24,25,26] and quantum wells [27,28]. A 3D geometrical model of the NW was built, on the basis on an appropriate mesh discretization, shown in Figure 1. Different regions were defined, in particular for the electrical contacts at the ends of the NW that were considered ohmic.

The presence of a thin Ge oxide was disregarded as it is nowadays recognized that oxide removal prior to functionalization is a necessary step in order to obtain efficient molecular sensing devices [3,12,29,30].

The semi-classical transport simulation of electrons and holes is based on the drift-diffusion approximation, based on the system of partial differential equations
(1)$$-\nabla \left(\varepsilon \nabla \varphi \right)=\;-e\left(n-p-N_{d}^{+}+N_{a}^{-}\right)\;-\nabla (\mu_{n}n\left(\nabla \;\phi_{n}\right)\;=\;R\;-\nabla (\mu_{p}p\left(\nabla \;\phi_{p}\right)\;=\;-\;R$$
where *N^+^_d_* and *N^−^_a_* are the densities of ionized donors and acceptors, *R* is the net recombination rate, *µ_n_* and *µ_p_* are mobilities for electrons and holes, *ϕ_n_* and *ϕ_p_* the Fermi energies for electrons and holes.

This system was solved for electrons and holes in 3D at room temperature, within a self-consistent approach that calculates the energy band system considering a bulk density of states *g*(*E*) and the presence of impurity levels and surface trap states in various regions of the NW.

The calculation allows to derive the value of Fermi energy values and thus, the carrier densities *n* and *p* due to the ionization of such states, following
(2)$$\begin{array}{c} n=\int g\;\left(E-E_{c}\right)f\left(E+q\phi_{n}\right)dE\\ p=\int g\;\left(E_{v}-E\right)\left[1-f\left(E+q\phi_{p}\right)\right]dE \end{array}$$

Recombination processes were taken in consideration, in particular SRH (Shockley-Read-Hall) processes, for which we assumed bulk Ge parameters. Radiative recombination processes were considered not relevant, due to the indirect gap nature of this material.

Realistic NW parameters such as length, size, composition were taken from characterization data from cited works, while material parameters such as lattice parameter, energy bands, effective mass were taken on the available literature data—except of values of mobility where the calculation on the basis of a bulk doping-dependent method [31] results in much higher values than those found experimentally, as evidenced by Zhang et al. [5]. Henceforth, when surface traps were included in the calculation, for this particular input parameter we considered a value of 300 cm^2^ V^−1^ s^−1^ that takes in consideration surface scattering due to such traps [5,6].

The calculation results allow for a complete three-dimensional picture of the NW properties; in particular, we were able to obtain energy bands, carrier densities, and conductivity values in all regions of the simulated structure, alongside the 3D current density vector and the current intensity at the contacts. Moreover, by applying a varying voltage bias at the end contacts of the NW, we could also calculate the relative I–V characteristic for the structure in consideration.

In order to validate the simulation of properties of Ge NWs modeled as discussed above, as a first step we carried on a calculation considering the nanostructures characterized in [32]. Hence, we calculated the I–V characteristic for a Ge NW with same morphological properties (95 nm diameter, 9.6 µm length) and with electrical contacts 2 µm long on both sides of the NW. Within the model, such contacts were considered to be ohmic and with a negligible resistance in comparison with the nanowire’s resistance.

Lacking experimental information on these input parameters, the values of doping (acceptor-like 2 × 10^17^ cm^−3^–0.16 eV above VB) and surface trap density (10^12^ cm^−2^–0.11 eV above VB) were taken from the literature [5,33].

In Figure 2 we present the result of the Tibercad calculation on the as-grown NW, in terms of energy band calculation, carrier density and majority carriers’ conductivity. In Figure 2a, we show the energy values for the peak of the valence band and the bottom of the conduction band as a function of the transverse axis going through the center of the NW, similarly to what was done in [27,34]. This plot shows the expected band bending on the surface, as an effect of acceptor-like surface traps. Due to these band profiles, majority carriers (being hole, as the NW is p-doped) accumulate on the surface (green line in Figure 2 right) and, as a consequence, cause an increase of conductivity in this area of the nanostructure, as shown in Figure 2 (left).

Hence, the carrier density varies from a value of 3.2 × 10^16^ cm^−3^ in the NW core to 2.5 × 10^17^ cm^−3^ in the NW outer shell, while the conductivity varies accordingly from 1.5 S/cm (core) to 12 S/cm (shell).

We run the same simulation for different voltage biases applied to the contacts at the end of the NW, aiming to derive the I–V characteristic in the range between 0 and 0.1 V. The results are shown in Figure 3, where they are plotted against the experimental data taken from [32].

The disagreement between experimental and modeled data is less than 10%, a remarkable value considering that in this modelization there are no free input parameters that can be adjusted to change the result of the simulation: morphological parameters were taken from experimental values, value of doping, surface traps and other material-specific parameters come from the best values available in literature. On this basis, we considered this result as a satisfactory validation of our model and, therefore, we deemed it suitable to study the mechanism of electric sensing of molecules.

## 3. Results and Discussion

As discussed above, three main physical effects can be considered when molecules interact with Ge NW, from the perspective of the influence on its electrical characteristic (i.e., a change in current induced by molecules):passivation of surface traps;additional surface charge;molecular charge transfer.

In the following, we consider separately these possible effects and quantitatively simulate the change in these parameters: (i) carrier density, (ii) conductivity, (iii) current at 0.1 V bias. If not expressly stated, all other parameters of the model were kept fixed and equal to the values stated above for the validation of the model.

### 3.1. Surface Traps Passivation

It has been argued that the effects of passivation of surface traps by deposition of molecules (in particular, isoprene monolayers) are twofold: (i) an increase of mobility thanks to the reduction of surface traps scattering [10]; (ii) a reduction of the effective hole density due to the reduction of the accumulation of holes due to the band bending at the NW surface [5]. To simulate this behavior, we removed the surface traps and modified accordingly the mobility to a value of 600 cm^2^ V^−1^ s^−1^, in agreement with literature data [5].

As evidenced in Figure 4, where we report the band profiles and the hole density along the transverse axis, the removal of surface traps causes the elimination of hole accumulation on the surface; hence, the hole density takes the value of 3.2 × 10^16^ cm^−3^ (corresponding to the value at the NW core in the case with surface traps of Figure 2). This change in spatial charge makes the NW spatially homogeneous for the electrical properties: values of *p*, hole conductivity *σ_h_* and hole resistivity *ρ_h_* in the case of complete passivation of surface traps correspond to the values of these parameter found in the core of the NW. The combination of reduced hole density and increased surface mobility results in a conductivity of 15 S/cm.

The simulation of the I–V characteristic produced a prediction of a current of 2.2 × 10^−8^ A with 0.1 V bias, a reduction of about 47% as compared to the situation with surface traps. Therefore, these calculations indicate that a total removal of surface traps causes only a halving of the current, due to the combination of two different effects: increase of mobility due to the elimination of surface trap scattering and reduction of hole density. It is important to notice that this should be considered as an upper limit, because surface traps were completely removed and, in realistic condition, it is more probable for molecules to passivate only a fraction of all surface traps.

### 3.2. Additional Surface Charges

Next, we consider the occurrence of additional charges accumulating on the surface of the NW due to the molecules adsorbtion: in order to evaluate the most extreme scenario we considered a high value of charge density of 10^12^ cm^−2^, that corresponds to maximum values commonly reported in literature [13]. In these simulations, surface traps were left active in the model.

To give a comprehensive picture we also considered the possibility of having molecules with negative and positive charges and run the simulation in these two opposite configurations.

Results of simulations are shown in Figure 5: when additional p-charge is added to the surface the band bending is reversed, resulting in higher values of carrier density and, hence, conductivity are now found in the core of the NW. Moreover, this drastic change in band profiles causes a strongly reduced hole density by almost on order of magnitude and a decreased conductivity (down to 0.45–1.5 S/cm). This behavior is in agreement with experimental works showing a reduced conductivity upon application of a positive external electric fields applied perpendicular to the NW surface by a gate electrode [10].

When a 0.1 V bias is applied, a net current of 5.4 × 10^−9^ A is calculated, meaning a reduction of about 87%. Such low values of current might raise some issue for signal detection and, hence, some doubts for the usefulness of these particular nanostructures for sensing molecules with positive charges.

On the other hand, adding negative charges results in an opposite scenario, as shown in Figure 6. The addition of negative charges enhances the band bending, resulting in an increase of hole accumulation on the NW surface, whose density reaches the value of 2.1 × 10^18^ cm^−3^, bringing the value of conductivity up to 100 S/m. This, in turn, results in a strong increase of the current caused by the application of a 0.1 V bias that reaches 1.58 × 10^−7^ A, almost three times the value of the situation with non-surface charges. It is interesting to notice that, in both cases of additional surface charges, the values of conductivity and carrier density in the core remain the same of the case of Figure 2, as there are actually no additional carriers added since the physical operation mechanism is the same of Field Effect Transistors.

It is noticeable how the sign of additional surface charges results in rather different outcomes in terms of change in electric signal for sensing.

### 3.3. Molecular Charge Transfer

The last effect due to interactions of molecules with Ge NWs is the transfer of charges from molecular orbitals to Ge bands. Although this is a process already observed and discussed in the literature, it is difficult to estimate a numerical value of the density of additional charges because the process depends on many variables such as temperature, molecule nature and details of molecule grafting on the NW.

Hence, for this case, we considered different values of additional doping, from a low level of from 5 × 10^16^ cm^−3^ to a maximum of 10^18^ cm^−3^ for both holes and electrons. Higher values of doping have been obtained only at temperatures higher than RT [15,18], so for the purpose of simulating electrical sensing of molecules at RT, we considered this maximum level of doping as a reasonable limit.

In Figure 7, we show the band profiles that result from such an extreme level of charge transfer: if holes with a density of 10^18^ cm^−3^ are transferred from the grafted molecule to the NW, the band bending is strongly reduced, making the system much more homogeneous from the core to the surface with the hole carrier density going from 6.3 × 10^17^ cm^−3^ in the core to 7.6 × 10^17^ cm^−3^ on the surfaces. Thanks to the high increase of carrier density (three times of the configuration of Figure 2), the application of a 0.1 V bias results in a current of 2.36 × 10^−7^ A, more than 4 times the value of Figure 2.

On the other hand, an electron charge transfer of 10^18^ cm^−3^ completely changes the electric properties of the system, as evidenced in Figure 7 (right panel). Due to the strong n-doping the majority carriers are now electrons and the band bending due to surface traps in reversed, resulting in a higher concentration of 6.5 × 10^17^ cm^−3^ carriers in the core of the NW. Hole density values (not shown) are reduced to 1.5 × 10^9^ cm^−3^ in the core to 1.5 × 10^11^ cm^−3^ on the surfaces, with a negligible contribution to the current.

Despite this lower value of carriers in comparison with hole charge transfer of same density, the current recorded upon bias applications is much higher (1.45 × 10^−6^ A against 2.36 × 10^−7^ A), due to the higher mobility of electrons. This strong increase of current is not unexpected, considering the rather high value of additional doping considered, a five-fold increase from the original p-doping value of 2 × 10^17^ cm^−3^. As discussed above, such situation could be very hard to achieve in a practical application, therefore we carried out more simulations at lower levels of molecular charge transfer to investigate intermediate values of additional doping and, also, to ascertain at what doping levels the holes switch to electron as majority carriers.

As evidenced in Figure 8, the transfer of holes from molecules to the NWs results in a monotonous increase of the current up to the value of 2.36 × 10^−7^ A. On the other hand, the situation is rather different if one considers electron transfer: in this case the current reduces strongly, down to few nA for n-doping above 10^17^ cm^−3^. Then, if the dopant density is increased beyond 6 10^17^ cm^−3^, the current (now due to electrons being majority carriers) increases up to 1.45 × 10^−6^ A.

It is interesting to study the carrier densities in more depth in the case of this additional n-doping. In Figure 9 we plot these values both in the core and on the surface of the NW.

As predictable, upon transfer of electrons from molecules, the electron and hole densities increase and reduce, respectively. However, due to the presence of acceptor-like surface states that causes the band bending, they accumulate in different areas of the NW: holes on the surfaces and electrons in the core. This causes a remarkable situation when additional dopant density of electrons is at 6 × 10^17^ cm^−3^: in the core the majority carriers are electrons, while on the surface they are holes. This causes the presence of two currents of different polarity in the two different regions; they compensate resulting in a very small current of 2.12 × 10^−9^ A measured at the contacts at the NW ends. Therefore, from a theoretical point of view a system could be realized where ambipolar transport of carriers occurs: this could be achieved by controlled additional n-doping (not necessarily by molecular charge transfer) of a p-doped NW with surface traps.

To summarize the findings of this modeling work, in Table 1 we show all relevant quantities for electrical parameters. The higher change in current (thus, the higher senstivity to molecules) could be achieved via a charge transfer mechanism, but only in the case of a very strong doping of 10^18^ cm^−3^, a condition that has not been reported so far for molecules grafting on NW at room temperature in standard conditions. Indeed, when the molecular doping is reduced to 5 × 10^16^ cm^−3^ the change in current is consistently decreased. On the other hand, it seems that the effect of fixed surface charge (that is, without any actual transfer of carriers into the NW) could allow for a more efficient electrical sensing; it should be noticed, however, that the sign of the charges has a great influence due to the acceptor-like surface states: we expect that, at same charge densities, the presence of electrons results in a change of current more than three times higher than for positive charges.

Obviously, it should be considered that—in a real situation and depending on the particular nature of the molecule—all these processes could be present at the same time and, possibly, combine and/or counterbalance their effects—for example, a charge transfer of electrons from a molecule could mean that positive ions remain on the surface, adding to the reduction of current. However, our calculations can be considered as an upper limit for the sensing mechanism under consideration. Thus, it can be argued that the sensing efficiency of a NW depends strongly not only on the NW properties but also on the nature of the molecule of interest and, in particular, on their interaction. In order to gain more insight on the molecule-nanowire interactions, first-principle calculations based on density functional theory can be used to investigate the effects of a particular molecule on the atomistic level. However, due to the much higher demands in terms of computing time and resources these approaches are better suited for modeling NW sensors targeting the sensing of a specific molecule [35,36].

## 4. Conclusions

In this work we have modeled the electrical properties of Ge NWs with the purpose of investigating mechanisms for electrical sensing of molecules. We have built a 3D model of a Ge NW with ohmic contacts at each end to solve the drift-diffusion equation, considering the presence of surface states. By using the software Tibercad to self-consistently solve the equation, we were able to obtain the energy band configuration, the 3D carrier densities and other electrical parameters and derive the electrical current. We have validated the model against experimental data from a previous publication and used it to consider different type of interactions between molecules and NW.

In particular, we considered: (i) the passivation of surface traps, (ii) the effect of fixed charges on the surface, (iii) the additional doping due to charge transfer.

By simulating these configurations, we quantitatively show that a charge transfer mechanism would allow for the stronger sensing of molecules, but only if a very high level of additional doping is achieved, so far not reported in the literature. On a more easily achievable situation, the higher change in current should result from the presence of negative charges on the surface. We also predict that, by having an additional n-type doping of 6 × 10^17^ cm^−3^, a situation with ambipolar transport could arise, with different majority carriers in the core of the NW (electrons) and on the surface (holes).

On a more general note, our calculations show that the electrical sensing mechanism can be very complex and it depends not only on the NW properties but also on the peculiar characteristics of the molecules to be sensed. We hope that this theoretical work can foster experimental verifications of these predictions and contribute to a deeper understanding of the sensing mechanisms in NWs, towards the final goal of developing more efficient molecular sensors.

## Figures and Tables

**Figure 1 nanomaterials-11-00507-f001:**
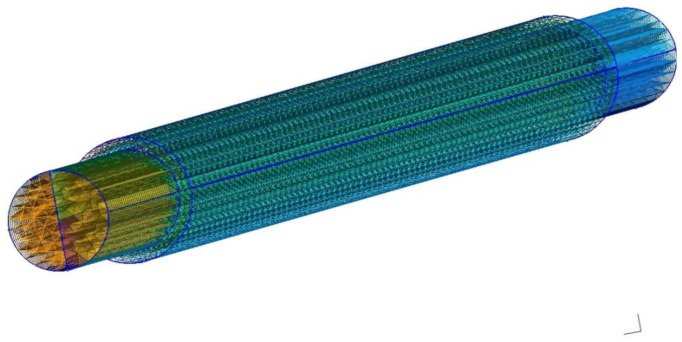
Image of the 3D model for the Ge NW. The nanowire is entirely made of germanium, with electrical contacts on the surface of front and back ends. Triangular shapes indicate the 3D mesh used for volume discretization.

**Figure 2 nanomaterials-11-00507-f002:**
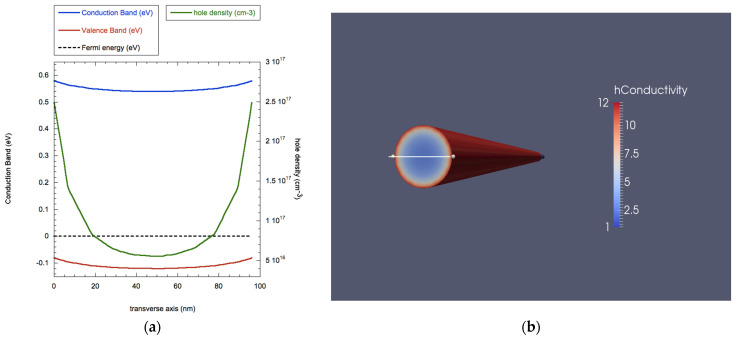
(**a**) Values for energy bands (blue: bottom of conduction band, red: peak of valence band), Fermi level (dashed black line) and carrier density (green line) along the trasverse axis; (**b**) transverse section of a 3D plot of conductivity for a Ge NW. The white line indicates the transverse axis for band profiles.

**Figure 3 nanomaterials-11-00507-f003:**
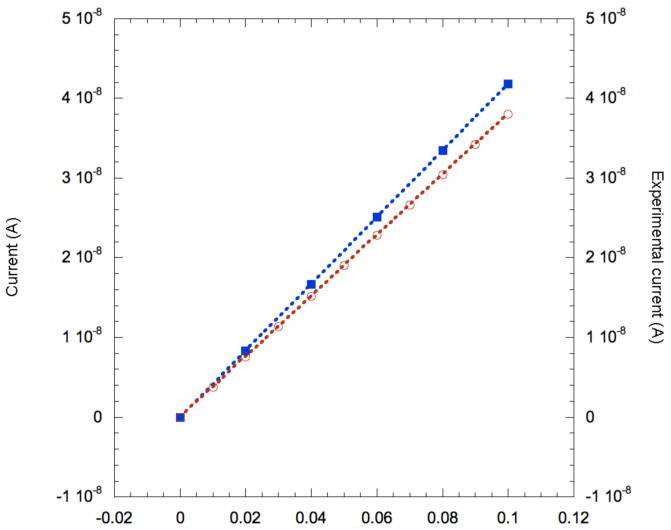
Comparison between I–V characteristics from Tibercad simulation (blue symbols) and experimental data (red symbols). Lines are guide for the eye.

**Figure 4 nanomaterials-11-00507-f004:**
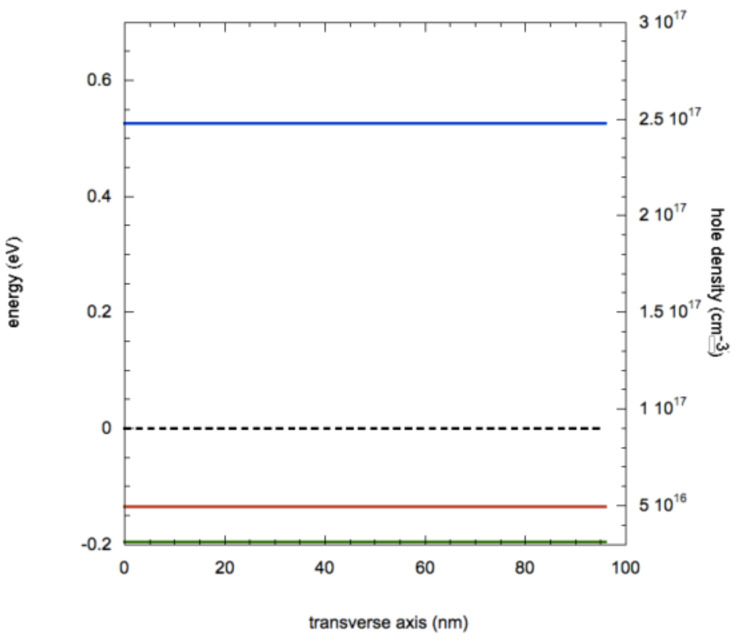
Values for energy bands (blue: bottom of conduction band, red: peak of valence band), Fermi level (dashed black line) and carrier density (green line) along the transverse axis for a Ge NW without surface traps.

**Figure 5 nanomaterials-11-00507-f005:**
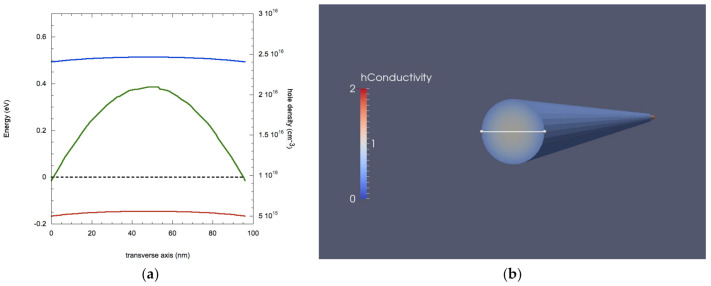
(**a**) Values for energy bands (blue: bottom of conduction band, red: peak of valence band), Fermi level (dashed black line), and carrier density (green line) along the transverse axis with additional p surface charges. (**b**) Transverse section of a 3D plot of conductivity.

**Figure 6 nanomaterials-11-00507-f006:**
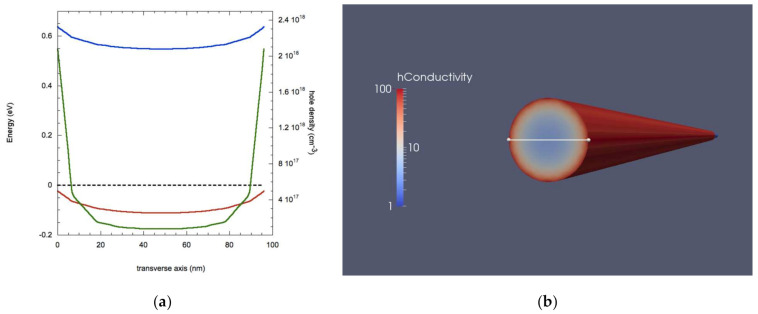
(**a**) Values for energy bands (blue: bottom of conduction band, red: peak of valence band), Fermi level (dashed black line) and carrier density (green line) along the transverse axis for a Ge NW with additional n surface charges. (**b**) transverse section of a 3D plot of conductivity.

**Figure 7 nanomaterials-11-00507-f007:**
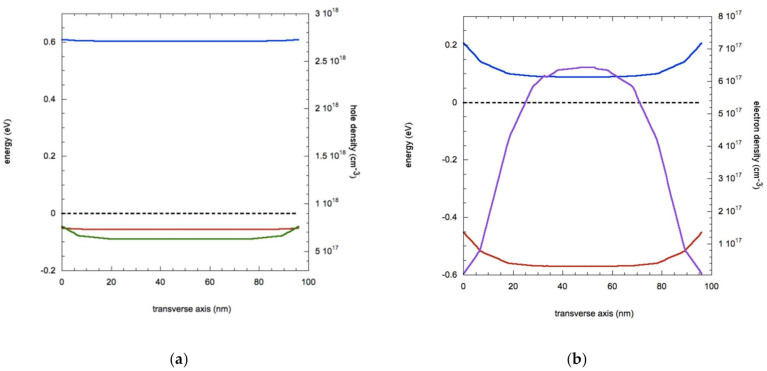
(**a**) Values for energy bands (blue: bottom of conduction band, red: peak of valence band), Fermi level (dashed black line) and hole carrier density (green line) along the transverse axis for a Ge NW with 10^18^ cm^−3^ hole charge transfer. (**b**) Band profiles (blue: conduction band, red: valence band), Fermi level (dashed black line) and electron carrier density (purple line) along the transverse axis for a Ge NW with 10^18^ cm^−3^ electron charge transfer.

**Figure 8 nanomaterials-11-00507-f008:**
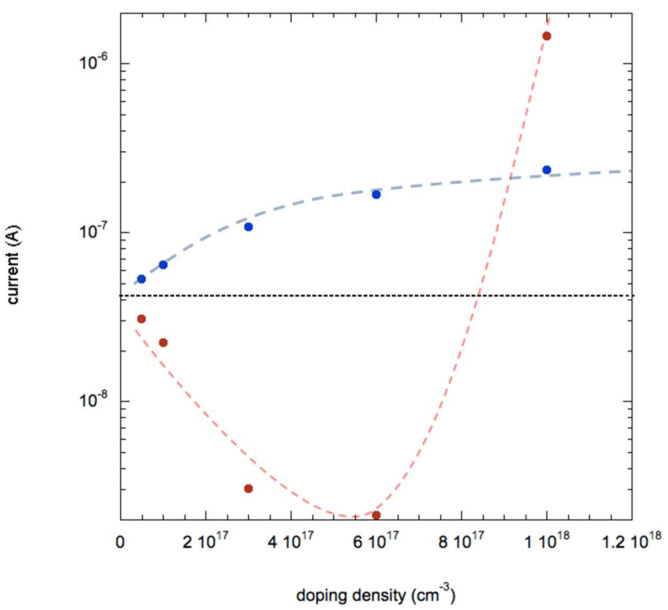
Current dependence on additional dopant density due to molecular charge transfer for electrons (n-doping, red symbols) and for holes (p-doping, blue symbols). The dotted black line indicates the value for a NW without charge transfer. Dashed lines are guide for the eye.

**Figure 9 nanomaterials-11-00507-f009:**
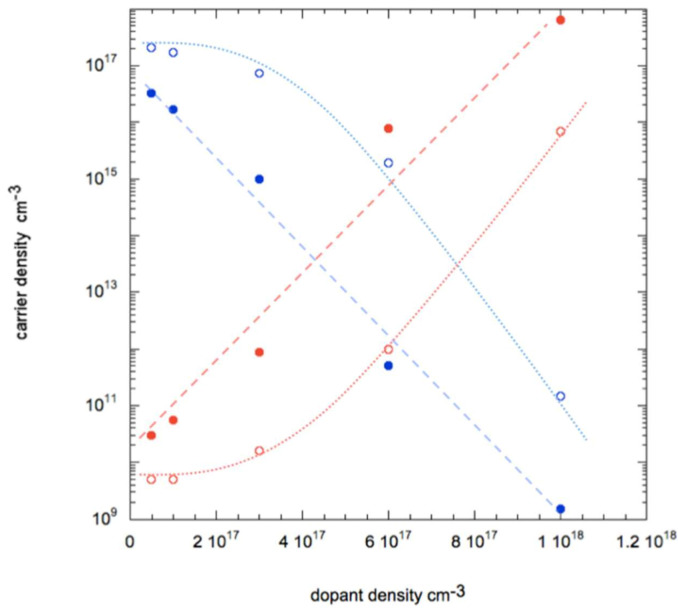
Carrier density plotted as function of density of transferred charge considering holes (blue open symbols) and electrons (red open symbols) on the surface on the NW and holes (blue filled symbols) and electrons (red filled symbols) in the core on the NW. Lines are guide for the eye.

**Table 1 nanomaterials-11-00507-t001:** Calculated parameters for different sensing processes considered.

Molecule Interaction	Current at 0.1 V Bias (A)	Carrier Density (Surface—Core) × 10^16^ cm^−3^	Conductivity (Surface—Core) S/cm	Resistivity (Surface—Core) ohm cm	Relative Change in Current (%)
none	4.18 × 10^−8^	25–3.2	12–1.5	0.08–0.67	-
surface traps passivation	2.2 × 10^−8^	3.2	1.5	0.67	47
fixed surface charge (n)	1.5 × 10^−7^	210–3.2	100–1.5	0.01–0.67	278
fixed surface charge (p)	5.4 × 10^−9^	0.9–3.2	0.45–1.5	2.22–0.67	−87
charge transfer (n)—5 × 10^16^ cm^−3^	3.09 × 10^−8^	21–3.2	10–1.5	0.1–0.67	−26
charge transfer (p)—5 × 10^16^ cm^−3^	5.30 × 10^−8^	28–8.6	13.6–4	0.07–0.25	27
charge transfer (n)—10^18^ cm^−3^	1.45 × 10^−6^	0.68–65 (electrons)	4.3–410 (electrons)	0.23–0.002 (electrons)	13,369
charge transfer (p)—10^18^ cm^−3^	2.36 × 10^−7^	76–63	37–30	0.027–0.033	464

Two values of carrier densities, conductivities and resistivities are indicated: on the outer surface shell and in the core of NW. If not indicated otherwise, values are considered for holes (majority carriers).

## Data Availability

The data presented in this study are available on request from the corresponding author. The data are not publicly available due to IP reasons.

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
