# Peer review of "Germanium Nanowires as Sensing Devices: Modelization of Electrical Properties"

_nanomaterials, 2021, doi:10.3390/nano11020507_

Round 1
Reviewer 1 Report
The manuscript presents results of very interesting research and is certainly of high relevance. However, I find that the presentation of results could be improved somewhat.
- By looking at Figure 1 it is absolutely not clear what the simulated structure is. Is the entire nanowire made of germanium with only front and back metallic contacts or is the germanium only a shell for the metallic nanowire? Also, why does the structure exhibit triangular shapes? The reader can deduce these by reading further, but explaining this in the caption of Figure 1 would really help a reader who is not familiar with the software that the authors use.
- In figures 2a, 4, 5a, 6a and 7, the authors show what they call "band profiles". It is generally accepted that a band profile is the depiction of the region of energies available for carriers in that band presented for certain points in the Brillouin Zone. What authors present on the aforementioned figures are (what I assume) the energy values for the peak of the valence band and the bottom of the conduction band as a function of a point on the transverse axis of the nanowire. This needs to be clarified.
- At the End of the introduction section there is a sentence from an article template that should be removed: "References should be numbered in order of appearance and indicated by a numeral or numerals in square brackets—e.g., [1] or [2,3], or [4–6]. See the end of the document for further details 80 on references. "
Minor issue - line 153 - "energy bad" should be corrected to "energy band"
Author Response
We thank the referee for pointing out flaws in our manuscript - we corrected them as it follows:
1 - Caption of Figure 1 has been modified to clarify that the nanowire is entirely made of germanium, with electrical contacts on the surface of front and back ends. The triangular shapes indicate the three-dimensional mesh we used for volume discretization to solve the equations. The new caption is:
" Figure 1. Image of the 3D model for the Ge NW. The nanowire is entirely made of germanium, with electrical contacts on the surface of front and back ends. Triangular shapes indicate the 3D mesh used for volume discretization."
2 - All captions of Figs. 2a, 4, 5a, 6a and 7 have been modified as "... Values for energy bands (blue: bottom of conduction band, red: peak of valence band)...".
Moreover, we changed the text on page 4 when presenting the plot of energy bands adding also a citation of a previous work where presented a similar plot: "In Fig.2 (a) we show the energy values for the peak of the valence band and the bottom of the conduction band as a function of the transverse axis going through the center of the NW, similarly to what was done in Refs. [27, 34].".
3- the sentence from the template was removed
4- We corrected the typing error of line 153
Reviewer 2 Report
In the present work the authors have studied the electrical properties of Germanium NWs focusing on the electrical sensing of molecules. By using the software Tibercad to solve the equation, they obtain the energy band configuration, the 3D carrier densities and other electrical parameters and derive the electrical current. Using already existing experimental data they consider different type of interactions between molecules and NW. In particular, they consider: i) the passivation of surface traps, ii) the effect of fixed charges on the surface, iii) the additional doping due to charge transfer. By simulating these configurations, they show that a charge transfer mechanism would allow for the stronger sensing of molecules, but only if a very high level of additional doping is achieved, so far not reported in the literature. On a more easily achievable situation, the higher change in current should result from the presence of negative charges on the surface. We also predict that, by having an additional n-type doping a situation with ambipolar transport could arise, with different majority carriers in the core of the NW (electrons) and on the surface (holes).
In general the paper is interesting, the analysis is well written and the results are supported by the data. The agreement between theory and experiment is fairly good. Therefore I accept for publication the paper in its present form.
Author Response
We thank the reviewer for the very positive judgement of our work.
Reviewer 3 Report
This is a well-written paper on and interesting topic. The results, such as displayed in Figure 3, are very good, particularly the close agreement with experiments.
While this simplistic semi-classical/empirical model is shown to be powerful, the authors might was to add a few sentences on how they might want to extend this to a more first-principles approach.
I need not see the next version of this paper unless it is significantly changed such as due to responses to other reviewers.
Author Response
We thank the referee for his positive evaluation of our work. His point on the extension of simulations towards first-principle methods is very relevant - indeed, when the interest is focused on sensing one particular molecule, atomistic approaches such as DFT or TB allow for a deeper analysis.
In this paper we used this classical method to give a general perspective on three possible processes for sensing different types of molecules, as our model is less demanding in terms of computational resources and time. Nevertheless, in our plans there is the study of sensing specific molecules by Ge NWs, so the suggestion of the referee is very valuable.
We added two sentences on this point:
- pag.2 - Introduction: "For this, we solve the classical drift-diffusion differential equations, by relying on empirical parameters provided by experimental data and by previously published results. This approach allows for simple and fast calculations of NW electrical properties considering different molecules and different interactions."
- pag. 12 - right before Conclusions: "In order to gain more insight on the molecule-nanowire interactions, first-principle calculations based on Density Functional Theory can be used to investigate the effects of a particular molecule on the atomistic level. However, due to the much higher demands in terms of computing time and resources these approaches are better suited for modelling NW sensors targeting the sensing of a specific molecule. [34,35]"